# Translation and Validation of the Richards–Campbell Sleep Questionnaire for Intensive Care Unit Patients in Morocco: Reliability and Validity Assessment

**DOI:** 10.3390/clockssleep7030031

**Published:** 2025-06-23

**Authors:** Abdelmajid Lkoul, Keltouma Oum’barek, Mohamed Amine Baba, Asmaa Jniene, Tarek Dendane

**Affiliations:** 1Laboratory of Biostatistics, Clinical Research and Epidemiology, Faculty of Medicine and Pharmacy, Mohammed V University, Rabat 10010, Morocco; abdelmajid_lkoul@um5.ac.ma (A.L.); oumbarekeltouma@gmail.com (K.O.); dendanetarek@gmail.com (T.D.); 2REGNE Laboratory, Faculty of Medicine and Pharmacy of Agadir, Ibn Zohr University, Agadir 80000, Morocco; 3Exercise Physiology and Autonomic Nervous System Team, Laboratory of Physiology, Faculty of Medicine and Pharmacy, Pulmonology Department, Ibn Sina Hospital, Ibn Sina University Hospital Center, Mohammed V University, Rabat 10010, Morocco; a.jniene@um5r.ac.ma

**Keywords:** sleep quality, sleep deprivation, intensive care unit, RCSQ, Moroccan version

## Abstract

Introduction: For patients in intensive care units, the Richards–Campbell Sleep Questionnaire (RCSQ) seems to be a useful tool for assessing sleep quality. However, its application in the Moroccan medical context could be limited due to the lack of a dialectal Arabic version for Morocco. This study’s objective was to translate and validate the RCSQ into Arabic for Moroccan speakers. Patients and methods: For this investigation, a cross-sectional design was adopted. The RCSQ was translated and validated into Arabic for Morocco in accordance with the recommendations. For every scale, psychometric properties were computed. The Cronbach’s α coefficient was utilized to evaluate the internal consistency of multi-item measures. Results: The study involved 224 patients, whose mean age was 47 ± 18.3 years. The RCSQ’s internal consistency, or Cronbach’s alpha, was computed, and all dimensions showed good reliability over the 0.92 (0.894–0.983) level. The items demonstrated good reliability and validity, with correlation values larger than 0.4, according to the data. Conclusion: The RCSQ translated into Arabic for Morocco appears to have good psychometric qualities, making it useful for assessing the quality of sleep of patients in intensive care units within Moroccan healthcare settings.

## 1. Introduction

Sleep is essential for physiological and cognitive recovery in humans, playing a crucial role in maintaining the body’s protective and immune functions [1]. Globally, sleep disorders are a major public health issue, leading to both individual and societal consequences, such as an increased risks of chronic diseases and psychiatric disorders [2,3,4]. Numerous studies have demonstrated the importance of sleep for overall health, longevity, and quality of life [5,6,7,8]. Among patients, particularly those in intensive care units (ICUs), sleep disorders are common and include partial or total sleep deprivation, sleep fragmentation, and alterations in sleep architecture, such as increased latency, reduced efficiency, and frequent awakenings [9,10,11]. These disturbances can persist after hospital discharge, potentially becoming chronic and hindering post-hospital recovery [12]. Environmental factors such as nighttime medical interventions, excessive noise, and constant lighting in ICUs contribute to sleep disruption [13,14]. Additionally, the severity of the illness, mechanical ventilation, sedative use, and stress further exacerbate these problems [15,16,17]. These sleep disturbances are associated with slower recovery, increased morbidity and mortality, and prolonged ICU stays [2,18,19]. Poor sleep quality is also linked to mood disorders, increasing the risk of delirium in ICU patients [20], as well as a heightened risk of long-term cognitive impairment after hospital discharge [21]. Early detection of sleep disorders in ICU patients is crucial for mitigating their effects. Direct methods such as polysomnography (PSG), the bispectral index, and actigraphy are considered gold standards for sleep evaluation, but their use in ICUs is limited due to the irregular sleep–wake cycles of patients and the need for continuous monitoring [22]. Furthermore, these methods are expensive and require specialized personnel for setup and interpretation, which can affect the reliability of results [23]. Given these constraints, indirect methods such as self-reported questionnaires are more practical in ICUs. The Richards–Campbell Sleep Questionnaire (RCSQ), developed by Richards [24], was specifically designed for ICUs and validated against PSG. It provides a quick assessment of patient-perceived sleep quality, measuring five key aspects: sleep depth, time to fall asleep, frequency of awakenings, time spent awake, and overall sleep quality [24]. The RCSQ uses a simple scoring system, ranging from 0 (poor quality) to 100 (optimal quality), making it an easy-to-use tool for clinicians [25]. Widely used in English-speaking countries, the RCSQ has also been adapted into several languages, including Japanese, Chinese, Spanish, Korean, Thai, and German [26,27,28,29,30,31]. Given the absence of a validated Moroccan Arabic version of the RCSQ, a language widely spoken in Morocco, it is crucial to develop an adapted version that considers the linguistic of the local context to enhance clinical assessment of sleep quality in patients within intensive care units (ICUs). Morocco, with its linguistic diversity, requires measurement tools tailored to its dialectal and population-specific characteristics to ensure reliable and relevant evaluations. By undertaking the translation of the RCSQ into Moroccan Arabic using standardized methods, our study addresses a significant need in Moroccan healthcare, where language barriers can compromise both the quality of care and the accuracy of clinical assessments.

## 2. Results

### 2.1. Sample Characteristics

The study sample included a total of 224 patients admitted to critical care. The average age of the participants was 49 years (±18.3), with a slight predominance of females, representing 52.24% of the sample. Most admissions to the intensive care unit were for medical reasons, accounting for 72.7% of cases. Among these, 26.9% were due to respiratory diseases, 44.2% were related to cardiac failure, and 23.1% were associated with metabolic disorders. The mean APACHE II score was recorded at 9.60 (±5.87), while the Charlson Comorbidity Index averaged 1.02 (±1.34). Patients reported an EVA score of 3.79 (±3.18). At the time of completing the questionnaire, the average duration of hospitalization was 5 days (4–6). Importantly, there were no missing data for any of the baseline or outcome measurements, including the AM-RCSQ. Detailed characteristics of the study sample are presented in Table 1.

The overall internal consistency of the scale was found to be excellent, demonstrated by a Cronbach’s α coefficient of 0.960 (95% confidence interval), indicating a high level of reliability in the measurement. Furthermore, the internal consistency remained strong even when each item was omitted individually, with Cronbach’s α values ranging from 0.946 to 0.969. This suggests that no single item significantly detracted from the overall reliability of the scale, thereby reinforcing the robust internal consistency of the AM-RCSQ across all items (Table 2).

#### Test–Retest Reliability (ICC)

The total AM-RCSQ score demonstrated outstanding reliability, with an intraclass correlation coefficient (ICC2.1 = 0.978 [95% CI: 0.936–0.992]). Reliability was similarly excellent across all individual scale items: sleep depth (ICC2.1 = 0.971), sleep latency (ICC2.1 = 0.960), number of awakenings (ICC2.1 = 0.938), ability to return to sleep (ICC2.1 = 0.967), and overall sleep quality (ICC2.1 = 0.950). These results indicate high test–retest consistency for both the total score and each individual item of the scale (Table 3).

### 2.2. Validity of AM-RCSQ

#### 2.2.1. Clinical Validity

The AM-RCSQ version has exhibited a robust correlation with the findings from studies employing the original RCSQ [14], thereby underscoring its validity and reliability. This strong alignment with established research enhances its credibility and reinforces its effectiveness as a clinical tool for assessing sleep quality in various patient populations. The demonstrated consistency in results indicates that the AM-RCSQ can be trusted to provide accurate insights into sleep quality, making it a valuable resource for clinicians in their practice.

The Table 4 shows that individuals with poor sleep quality are significantly older (51.7 ± 17.7 years) than those with good sleep quality (40.1 ± 17.4 years), with a *p*-value < 0.001. Pain levels are also higher in the poor sleep quality group (4.44 ± 3.12) compared to the good sleep quality group (1.78 ± 2.57), with a *p*-value < 0.001. Anxiety is more common in the poor sleep quality group (35/145) than in the good sleep quality group (2/42), with a *p*-value of 0.017. Gender did not show a significant difference (*p* = 0.315). These findings confirm that age, pain, and anxiety significantly impact sleep quality, while gender does not. Age, pain, and anxiety are well-established factors associated with poor sleep quality, as confirmed by various scales, including the AM-RCSQ, which demonstrates its clinical validity.(Table 4). 

#### 2.2.2. Construct Validity

The correlation coefficients between the AM-RCSQ items and their corresponding variables on the PSQI were as follows: −0.588 for sleep depth, −0.583 for sleep latency, −0.551 for awakenings, −0.554 for the ability to return to sleep, and −0.598 for sleep quality. Additionally, the correlation between the total AM-RCSQ score and the PSQI total score was −0.635. All these correlation coefficients were statistically significant at the 0.01 level, indicating a strong inverse relationship between the AM-RCSQ scores and the PSQI variables, thereby strongly supporting the scale’s construct validity. The results supporting the construct validity of the AM-RCSQ are presented in Table 5.

#### 2.2.3. Discriminant Validity

The mean AM-RCSQ score of 224 patients was 34.8 ± 24. All components of the AM-RCSQ in the subjects with the highest quartile were higher than those of the lowest quartile (*p* < 0.001).

### 2.3. Factor Structure of AM-RCSQ

#### Exploratory Factor Analysis of AM-RCSQ

An analysis of skewness values for 224 patients revealed that all items followed a normal distribution (*p* > 0.05). The factor analysis of the AM-RCSQ yielded a KMO index of 0.937, and Bartlett’s test of sphericity produced χ^2^ = 1533, df = 15, and *p* < 0.001, indicating that the data were suitable for factor analysis. The principal component analysis identified a single factor accounting for 84.4% of the total variance. As shown in Table 6, all item factor loadings exceeded the established threshold, confirming the factor model’s robustness.

## 3. Methods

### 3.1. Study Design

A prospective cross-sectional study was carried out between September 2023 and August 2024, including 224 critically ill patients from three ICUs in three hospitals in the Souss-Massa region in southern Morocco. This design was selected to address the study’s key objective: assessing the reliability of the Richards–Campbell Sleep Questionnaire (RCSQ). The sample size was determined based on established recommendations, which indicate that at least 50 patients are needed to sufficiently evaluate the internal consistency of scales and assessment instruments [32].

### 3.2. Ethical Issues

This research protocol was conducted in accordance with the World Medical Association Declaration of Helsinki [33]. Prior to its commencement, the protocol received formal approval from the Ethics Committee for Biomedical Research, Faculty of Medicine and Pharmacy, Rabat, Morocco (approval No. 85/24). All participants were fully informed about the study’s objectives and procedures and provided written informed consent before their inclusion in the study.

### 3.3. Translation of the RCSQ

Dr. Richards, the original developer of the RCSQ, was consulted, and permission for the translation of the questionnaire was granted. The Arabic for Morocco version (AM-RCSQ) was developed in accordance with established guidelines [34,35]. Initially, two healthcare professionals specializing in intensive care, both fluent in English, independently translated the RCSQ from English to Moroccan Arabic. These two versions were then compared and consolidated by an ICU expert with over 10 years of experience in critical care. A back-translation into English was subsequently performed by a linguist and a sleep specialist, both fluent in Arabic and English. A panel of seven bilingual healthcare professionals, all actively involved in the study, reviewed the six versions of the AM-RCSQ. The translation process continued until all discrepancies were resolved, ensuring precision and accuracy. Finally, cognitive debriefing was conducted with four participants (three ICU physicians and one sleep specialist) to verify that each question of the AM-RCSQ was clearly understood by all participants. The degree of convergence of the experts’ answers during the final validation was measured by the Aiken coefficient, which showed a very high level of V = 0.82. The validated version of the AM-RCSQ was tested on 12 intensive care patients. Participants reported no ambiguity in their understanding of the questions. The final version of the AM-RCSQ was not re-adjusted after testing (Figure 1). 

### 3.4. Setting and Participants

This study took place in the ICU of Hassan II University Hospital in Agadir and the prefectural hospital of Taroudant and Tiznite. Participants were approached between 9 a.m. and 10 a.m. by one of two trained nurses in the ICU who assessed their eligibility and enrolled them using a non-probabilistic consecutive sampling method. Eligibility criteria for ICU patients included being 18 years or older, having a Glasgow Coma Scale score of 15, maintaining hemodynamic stability, and spending at least one full night (10 p.m. to 6 a.m.) in the ICU. Once enrolled, participants underwent a medical history review, followed by a clinical examination. integrated as part of the standard clinical evaluation. Sample size calculations indicated that a minimum sample of 50 patients was required to ensure 80% power, 5% significance, and 0.50 effect size [24].

### 3.5. Patient-Reported Outcome: The Richards–Campbell Sleep Questionnaire (RCSQ)

The RCSQ is a concise, self-administered questionnaire designed for ICU patients, consisting of five key items that assess different aspects of sleep: sleep depth, time to fall asleep, awakenings, return to sleep, and overall sleep quality. An optional item evaluating ICU noise is also included and rated separately. The questionnaire specifically measures the patient’s sleep from the previous night, using a 100 mm visual analog scale. Scores are determined by measuring the distance (in mm) from the lower end of the scale to the patient’s mark for each item. The final score is calculated by averaging the item scores, ranging from 0 (the worst possible sleep) to 100 (the best possible sleep).The Richard Campbell Sleep Questionnaire (RCSQ) uses a visual analog scale (0–100 mm) for five sleep domains; a total score < 50 indicates poor perceived sleep quality, whereas scores ≥ 70 reflect relatively good sleep [24].

### 3.6. Data Collection

Data collection for the translated version of the AM-RCSQ were conducted by a trained nurse who visited each unit during working shifts. To minimize recall bias and ensure accurate recollection of the previous night’s sleep, assessments were conducted only once, between 9 and 11 a.m. Patients were asked to evaluate their sleep from the previous night, with each question being read aloud to them. After each question, patients were asked to mark their response on a paper-based visual analog scale, consisting of a 100 mm undivided line (mm indicating the poorest sleep, 100 mm indicating the best sleep). For patients who were unable to mark the scale themselves, they pointed to their chosen spot with a finger, and the investigator marked the scale accordingly.

To verify the construct validity of the AM-RCSQ, the Arabic version of the Pittsburgh Sleep Quality Index (PSQI) was administered to patients concurrently [36]. The PSQI is a self-report questionnaire designed to measure sleep quality. It has demonstrated reliability, with an internal consistency (Cronbach’s alpha) above 0.80 in various studies. The PSQI is composed of seven components: subjective sleep quality, sleep latency (time to fall asleep), sleep duration, habitual sleep efficiency, sleep disturbances, use of sleep medication, and daytime dysfunction. Each component is rated on a scale from 0 to 3 for individual items. The total PSQI score ranges from 0 to 21, with a score above 5 indicating significant sleep disturbances.

The Charlson Comorbidity Index is a clinical score used to assess a patient’s life prognosis based on the comorbidities they suffer from. It ranges from 0 upward, depending on the number and severity of comorbid conditions; a score of 0 predicts a 98% ten-year survival, whereas a score ≥ 5 suggests a significantly reduced survival rate.

### 3.7. Test–Retest Reliability of the AM-RCSQ

The AM-RCSQ was administered by independent raters to a subset of patients to evaluate its test–retest reliability. The two assessments were conducted independently, with each rater blinded to the other’s results, and separated by a 6 h interval. This interval was selected to minimize recall bias and ensure accurate recollection of the previous night’s sleep, as sleep in ICU patients tends to be fragmented. A longer time gap between assessments could have impacted patients’ perception of sleep quality. Due to incomplete responses during the pre-test phase, and to ensure reliable test–retest assessment, nurses were instructed to read the questions aloud to the patients.

### 3.8. Statistical Analysis

Data analyses were conducted by an author not involved in the previous phases using Excel (Microsoft) and JAMOVI software version 2.3.28 https://www.jamovi.org/download.html (accessed on 24 February 2024), respectively, considering statistical evidence of significance at 0.05. Each questionnaire item showed non-normal distribution (all *p* < 0.001). Descriptive analysis is summarized using mean and standard deviation (mean ± SD) or absolute and relative frequencies (n, (%)), depending on the variable type.

The internal consistency of the final version for the compound score of the AM-RCSQ was estimated using Cronbach’s alpha along with a 95% confidence interval (95%CI); Cronbach’s alpha was also calculated if-item-omitted [37]. Cronbach’s values were considered as acceptable (0.70 to 0.79), good (0.80 to 0.89), or perfect (>0.9) [38].

Test–retest reliability alongside its 95% confidence interval (95%CI) was estimated based on a single measurement, absolute agreement, two-way random effect model (ICC2.1) [39]. ICC values were considered unacceptable (<0.40), acceptable (0.40 ICC 0.75), or excellent (>0.75) [39]. The Kaiser–Myer–Olkin (KMO) index (>0.5) was used to determine the factorability of the data.

## 4. Discussion

This study aimed to translate and validate the original English version of the RCSQ into Arabic for Morocco (AM-RCSQ). It focused on ensuring semantic and conceptual equivalence. Additionally, the study evaluated the internal consistency, test–retest reliability, and overall measurement of the scale among ICU inpatients. The findings of this study suggest that the AM-RCSQ is a valid and reliable tool for assessing sleep in ICU patients in Morocco.

A comparison between two Moroccan translations reveals no notable differences in translation, which allows for generalization of the internal consistency results from the previous study [27,29,31,40]. Nevertheless, the current study provides evidence of content and construct validity for the AM-RCSQ without the need to remove any items, indicating that the scale is linguistically suitable for use with ICU patients in the Moroccan intensive care context.

The internal consistency analysis revealed a Cronbach’s alpha of 0.960, meeting the criteria for internal consistency with a value exceeding 0.7, which signifies acceptable reliability [41]. The removal of individual items did not significantly enhance the internal consistency, indicating a high degree of reliability and minimal redundancy. The original validation of the RCSQ was conducted by Richards et al. [24], with a Cronbach’s alpha of 0.90 in the English version. The seems to be consistent across various languages, with Cronbach’s a value > 0.80 [27,29,31,41]. In our study, involving 224 ICU patients, along with other translated versions, the AM-RCSQ proved to be a valid tool for assessing sleep quality in critically ill patients.

The corrected item total correlations of the RCSQ were all above the minimum acceptable threshold of 0.4 [42], which indicates internal homogeneity of the scale. Regarding validity, the study provided robust evidence supporting both content and construct validity for the Moroccan version of the RCSQ (AM-RCSQ). The results demonstrated that the scale was linguistically appropriate to the Moroccan intensive care context. During the translation process, none of the items required substantial changes to fit the Moroccan cultural context. Equivalence testing showed that more than 98% of healthcare experts rated all items as appropriately translated, and the overall content validity was rated as very high. Additionally, the AM-RCSQ was effective in distinguishing between good and poor sleepers in the ICU. Regarding construct validity, the correlations between the AM-RCSQ and the PSQI were consistently above 0.50, which supports the convergent validity of the AM-RCSQ when compared to other established sleep questionnaires. However, due to the lack of available data, the discriminant and convergent validity of the AM-RCSQ were not directly compared with other studies.

Our study demonstrated excellent test–retest reliability for the AM-RCSQ (ICC2.1 = 0.978, 95% CI = [0.936–0.992]), obtained in a non-simultaneous and blinded manner. This result also highlights that repeated assessments are reliable, allowing for consistent daily evaluations during hospitalization by different raters. These results are in line with those from a recent study by Varella et al. [28]. The differences in sample populations may explain the variation in findings between our study and previous research.

The exploratory factor analysis (EFA) further confirmed the presence of a single-factor structure, which aligns with the findings of the original RCSQ validation by Richards et al. [24]. This supports the notion that the RCSQ is a unidimensional tool, designed to accurately assess sleep quality in ICU patients. In this study, the cumulative variance explained by the factor was 84.4%, reflecting strong explanatory power for the concept of sleep in ICU patients. This variance is notably higher than the 72.2% reported in the original RCSQ validation study [24]. The difference in variance between the studies could be attributed to cultural differences in symptom expression and the diversity in patient populations. While the original study predominantly involved male patients from a medical ICU, our study included a more heterogeneous sample from various ICU settings.

The study’s main strengths include strict adherence to international guidelines for each phase of adaptation and reliability analysis, the fulfillment of a priori sample size calculations for each study phase and full reporting of consistency and reliability for each item in the questionnaire [35,42]. It is notable that many adaptations of the RCSQ, such as the Korean, Japanese, and Thai versions [27,30,31], did not fully describe their translation process or the corresponding statistical analysis.

In this study, it was not possible to validate the AM-RCSQ against a direct sleep assessment method. However, previous research has demonstrated fair to moderate correlations between the RCSQ and polysomnography [30], providing some confidence in its validity.

In summary, the AM-RCSQ is a valid and reliable instrument for assessing sleep in ICU patients in Moroccan patients. The RCSQ is already recommended by clinical practice guidelines for assessing sleep disturbances in ICU inpatients [43], and emerging evidence supports its clinical utility, including its responsiveness to sleep hygiene interventions and ICU environmental modifications. Using the AM-RCSQ for ICU patients could help develop strategies to optimize sleep and reduce the comorbidities associated with poor sleep quality or sleep deprivation in the critically ill population.

The adapted Richards–Campbell Sleep Questionnaire may also be utilized in various clinical settings beyond the intensive care unit, including general medical wards, palliative care, and outpatient consultations, thereby providing valuable assessments of patients’ sleep quality in diverse situations.

## 5. Conclusions

Critically ill patients generally suffer from poor sleep, which has a detrimental effect on their health outcomes. Consequently, interventions designed to promote sleep and improve health outcomes in this patient group should be rigorously tested in large-scale trials across a variety of ICU settings. The Moroccan version of the RCSQ has been validated as a reliable and appropriate instrument for measuring sleep quality in ICU patients, making it an effective tool for future research and clinical practice in Moroccan ICU context. The application of the Moroccan version of the RCSQ could facilitate more accurate, assessments of sleep quality, enabling targeted interventions to enhance patient comfort and recovery in Moroccan intensive care units.

### Strength and Limitations

The researchers implemented several strategies to minimize bias in this study. The translation process strictly followed established protocols and underwent multiple rounds of expert review, significantly enhancing the validity of the translated version [44]. Data collection was conducted exclusively by one investigator, thereby reducing the likelihood of performance bias. The heterogeneity of the sample’s diagnoses improved the generalizability of the findings and reduced the risk of selection bias. The absence of neurosurgical and neurological patients in the sample represents a possible limitation, as their sleep perception may differ, although this potential bias was mitigated through strict inclusion and exclusion criteria.

It is important to note that the Pittsburgh Sleep Quality Index (PSQI) is designed to measure subjective sleep quality over a one-month period, which may exceed the typical duration of a patient’s stay in the intensive care unit (ICU). This discrepancy could impact the relevance of the PSQI findings in the context of acute care settings.

Future research should consider recruiting a larger sample across multiple centers to provide deeper insights into correlations between patient demographics, other clinical characteristics, sleep quality perceptions, and the underlying factors contributing to poor sleep in critically ill patients.

## Figures and Tables

**Figure 1 clockssleep-07-00031-f001:**
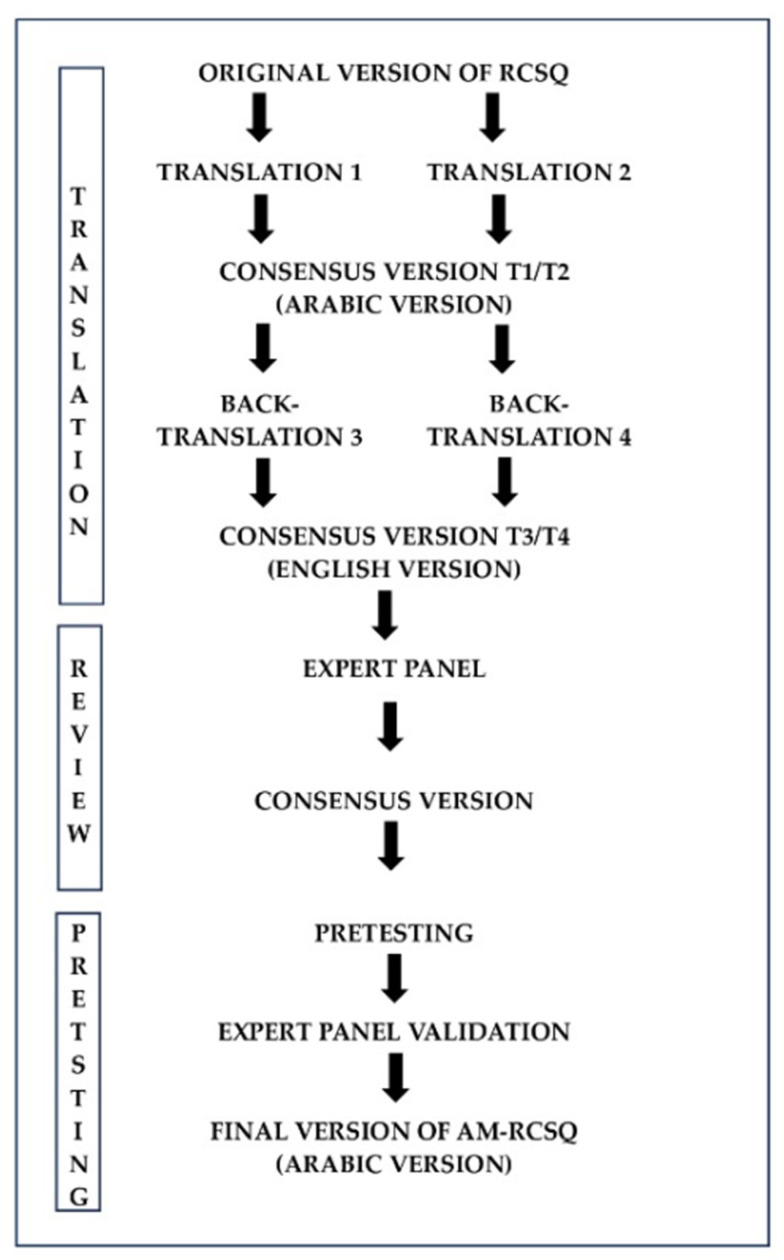
The process of translating and adapting the AM-RCSQ.

**Table 1 clockssleep-07-00031-t001:** Demographic and clinical characteristics of participants.

Characteristics	Total (n = 224)
Age, mean ± SD	49 ± 18.3
Gender, n (%) - Male/Female	107 (47.76)/117 (52.24)
Body mass index, kg/m^2^	23 ± 4.56
Admitting reason - Surgical/Medical	71(27.3)/189 (72.7)
Admission diagnosis, n (%) - Respiratory diseases - Heart disease - Metabolic disorders - Other medical reasons	70 (26.9) 115 (44.2) 60 (23.1) 15 (5.8)
Smokers	52(20)
APACHE-II scores, mean ± SD	9.60 ± 5.87
Charlson comorbidities score mean ± SD	1.02 ± 1.34
EVA-score, mean ± SD	3.79 ± 3.18
ICU length of stay (days), med (IQR)	5 (4–6)

**Table 2 clockssleep-07-00031-t002:** RCSQ: Richards–Campbell Sleep Questionnaire, content validity analysis (n = 224).

RCSQ Items	Descriptive		Internal Consistency
	Mean ± SD	Range	Cronbach’s α (If Item Omitted)
Sleep depth	36 ± 27.8	0–90	0.946
Sleep latency	32.7 ± 26	0–80	0.969
Awakenings from sleep	32.7 ± 25.8	0–80	0.967
Ability to return to sleep	33.8 ± 26.2	0–90	0.969
Sleep quality	37 ± 21.6	0–80	0.968
Item Optional: Noise	34.3 ± 19.1	0–80	0.969
Total score RCSQ	34.8 ± 24	0–80	0.960

SD: standard deviation.

**Table 3 clockssleep-07-00031-t003:** Test–retest reliability and measurement error analysis (n = 15).

	Test	Retest	Test–Retest Reliability	
	Mean ± SD (CV)	Mean ± SD (CV)	ICC2.1	95%CI
Sleep depth	56.7 ± 25.8	55.3 ± 24.8	0.971	0.919–0.990
Sleep latency	48.3 ± 29.1	52 ± 20.1	0.960	0.889–0.986
Number of awakenings	48.3 ± 22.1	48 ± 28.1	0.938	0.830–0.978
Ability to return to sleep	53.3 ± 28.1	47.3 ± 18.5	0.967	0.909–0.989
Overall sleep quality	53.3 ± 22.9	51.7 ± 26.3	0.950	0.861–0.983
Total RCSQ score (Average across the 5 items above)	51.4 ± 18.2	50.9 ± 17.3	0.978	0.936–0.992

SD: standard deviation.—ICC: Intraclass Correlation Coefficient—CI: Confidence Interval.

**Table 4 clockssleep-07-00031-t004:** Clinical validity of AM-RCSQ.

Variable	RCSQ		*p*-Value AM-RCSQ Version
	Good Sleep Quality	Poor Sleep Quality	
Age	40.1(17.4)	51.7(17.7)	<0.001 **
Pain mean ± SD	1.78(2.57)	4.44(3.12)	<0.001 **
Gender: M/W	24/20	83/97	0.315 *
Anxiety: Yes/No	2/42	35/145	0.017 *

* Chi-squared test, ** Student’s *t*-test.

**Table 5 clockssleep-07-00031-t005:** Construct validity of AM-RCSQ.

	Sleep Depth	Sleep Latency	Number of Awakenings	Returning to Sleep	Sleep Quality	Noise	Total RCSQ
PQSI	r: −0.588	r: −0.583	r: −0.551	r: −0.554	r: −0.598	r: −0.537	r: −0.635
	*p*: <0.001	*p*: <0.001	*p*: <0.001	*p*: <0.001	*p*: <0.001	*p*: <0.001	*p*: <0.001

PQSI: Pittsburgh quality sleep index, r: Spearman’s correlation, *p*: *p*-value.

**Table 6 clockssleep-07-00031-t006:** Exploratory factor analysis of AM-RCSQ.

	PCA **	MSA *
General	NA	0.937
Sleep depth	0.953	0.898
Sleep latency	0.906	0.955
Awakenings from sleep	0.921	0.930
Ability to return to sleep	0.900	0.955
Sleep Quality	0.911	0.948
Item Optional: Noise	0.920	0.947

ACP: Principal Component Analysis, MSA: Measure of Sampling Adequacy, * Kaiser–Meyer–Olkin = 0.937, Bartlett’s test of sphericity: χ^2^ = 1533, df = 15, *p* < 0.001 ** varimax rotation.

## Data Availability

The datasets generated and analyzed during the current study are available from the corresponding author on reasonable request.

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
