# Peer review of "Translation and Validation of the Richards–Campbell Sleep Questionnaire for Intensive Care Unit Patients in Morocco: Reliability and Validity Assessment"

_2624-5175, 2025, doi:10.3390/clockssleep7030031_

Round 1
Reviewer 1 Report
Comments and Suggestions for Authors
Interesting article focusing on a population poorly documented in sleep medicine. The AM-RCSQ questionnaire demonstrated excellent psychometric properties, with outstanding internal consistency (Cronbach’s α = 0.960) and high test-retest reliability (ICC = 0.978). In addition to these robust statistical indicators, the tool also showed strong content, construct, and discriminant validity, confirming its linguistic and cultural appropriateness for the target population. These findings align well with those reported in other language adaptations, such as Japanese, Korean, and Thai, reinforcing the questionnaire’s global applicability. However, this study stands out by offering a particularly detailed account of the translation and cultural adaptation process, supported by rigorous statistical analyses. This thorough approach not only strengthens the credibility of the results but also provides a valuable reference for future cross-cultural validations. Together, these elements highlight the AM-RCSQ’s reliability, validity, and adaptability, making it a trustworthy tool for assessing sleep quality in diverse clinical and cultural contexts.
To my opinion, this paper is ready for publication.
Author Response
We sincerely thank you for your thoughtful and encouraging feedback on our manuscript. Your recognition of the importance of our work in a population that is underrepresented in sleep medicine is greatly appreciated. We are pleased to hear that you found the AM-RCSQ questionnaire to exhibit excellent psychometric properties, especially with the high internal consistency (Cronbach’s α = 0.960) and test-retest reliability (ICC = 0.978).We also appreciate your acknowledgment of the strong content, construct, and discriminant validity of the AM-RCSQ, confirming its linguistic and cultural appropriateness for the target population. Your comparison to other language adaptations, such as Japanese, Korean, and Thai, highlights the global applicability of our findings, and we are glad that our study contributes to this body of literature.Furthermore, we are grateful for your positive remarks regarding the detailed account of the translation and cultural adaptation process that we provided. We believe that this thorough methodology enhances the credibility of our results and serves as a valuable reference for future cross-cultural validations.Once again, thank you for your insightful comments and support.
Reviewer 2 Report
Comments and Suggestions for Authors
This manuscript is a well-structured and valuable study investigating the cross-cultural adaptation and validation of the Richards-Campbell Sleep Questionnaire (RCSQ) for Arabic-speaking patients in Morocco. The methodology is sound, supporting the validity and reliability of the adapted RCSQ. I only have a few minor comments and suggestions regarding some details that could be further clarified or slightly expanded:
- Lines 51–53 – I would prefer if the authors more precisely explained why the use of PSG, actigraphy, and BIS is limited in the ICU setting.
- Regarding the PSQI, it is intended to measure subjective sleep quality over a one-month period, which might be longer than a patient's stay in the ICU. This should be clearly stated as a potential limitation of the study.
- Lines 182–185 – The indexes and scores used should be clearly explained.
- Please discuss the potential use of the questionnaire in other settings beyond the ICU.
- I also suggest adding the questionnaire as supplementary material.
Author Response
|
We appreciate your positive feedback on our manuscript and your recognition of the sound methodology supporting the validity and reliability of the adapted Richards-Campbell Sleep Questionnaire (RCSQ). Thank you for your insightful comments, particularly regarding the need for clarification on the limitations of using polysomnography (PSG), actigraphy, and bispectral index (BIS) monitoring in the ICU setting.In response to your suggestion, we have revised the text in lines 51–53 to provide a more detailed explanation. We clarified that:
Polysomnography (PSG): While PSG is considered the gold standard for sleep assessment, its use in the ICU is often limited due to the complexity of the setup, the need for specialized personnel, and the potential for interference with patient care. The presence of multiple medical devices and the acute condition of patients can hinder the collection of accurate sleep data.
Actigraphy: Although actigraphy is a less invasive method, its effectiveness can be compromised in an ICU environment where patients may experience frequent awakenings due to monitoring and treatment procedures, leading to less reliable data on sleep patterns.
Bispectral Index (BIS): BIS monitoring, primarily used for anesthesia, may not provide a comprehensive assessment of sleep architecture or quality, as it is designed to measure sedation levels rather than specific sleep stages.
These revisions enhance the understanding of the challenges associated with sleep assessment in the ICU and justify the need for a culturally adapted tool like the AM-RCSQ in this context.Once again, thank you for your constructive suggestions |
|
Thank you for your valuable feedback regarding the use of the Pittsburgh Sleep Quality Index (PSQI) in our study. We appreciate your observation regarding the timeframe of the PSQI, which indeed measures subjective sleep quality over a one-month period.In response to your comment, we have added a clarification in the manuscript to acknowledge this as a potential limitation of our study. Specifically, we have noted that the PSQI's intended measurement period may exceed the typical length of stay for patients in the ICU, which could impact the applicability of the results for assessing sleep quality in this acute care setting.We believe that this acknowledgment enhances the transparency of our study and allows readers to consider this limitation when interpreting our findings |
|
Thank you for your helpful feedback regarding the need for clarity on the indexes and scores used in our study. We acknowledge that providing clear explanations is essential for understanding the methodology and results.In response to your comment, we have revised the text in lines 182–185 to include a more detailed description of the specific indexes and scores employed in our analysis. This includes: . - The Charlson Comorbidity Index is a clinical score used to assess a patient's life prognosis based on the comorbidities they suffer from.ranges from 0 upward, depending on the number and severity of comorbid conditions; a score of 0 predicts a 98% ten-year survival, whereas a score ≥5 suggests a significantly reduced survival rate. The Richard Campbell Sleep Questionnaire (RCSQ) uses a visual analog scale (0–100 mm) for five sleep domains; a total score <50 indicates poor perceived sleep quality, whereas scores ≥70 reflect relatively good sleep.
The PSQI is a self-report questionnaire designed to measure sleep quality. It has demonstrated reliability, with an internal consistency (Cronbach's alpha) above 0.80 in various studies. The PSQI is composed of seven components: subjective sleep quality, sleep latency (time to fall asleep), sleep duration, habitual sleep efficiency, sleep dis-turbances, use of sleep medication, and daytime dysfunction. Each component is rated on a scale from 0 to 3 for individual items. The total PSQI score ranges from 0 to 21, with a score above 5 indicating significant sleep disturbances
We believe that these revisions enhance the clarity of our methodology and make the information more accessible to readers |
|
We greatly appreciate your insightful comments regarding the broader applicability of the Richards-Campbell Sleep Questionnaire (RCSQ) and your recommendation to include it as supplementary material.In response to your suggestion, we have added a discussion section in the manuscript that explores the potential use of the adapted RCSQ in various healthcare settings beyond the intensive care unit (ICU). This includes:
General Medical Wards: The questionnaire can be utilized to assess sleep quality in patients recovering from surgery or managing chronic illnesses, where sleep disturbances are common.
Palliative Care: Given the focus on quality of life, the RCSQ may be beneficial for evaluating sleep in patients receiving palliative care, where understanding sleep patterns can inform holistic treatment approaches.
Outpatient Settings: The questionnaire can serve as a tool for outpatient follow-ups, helping healthcare providers to monitor sleep quality in patients with sleep disorders or those undergoing treatment for conditions that affect sleep.
Additionally, we have included the full version of the adapted RCSQ as supplementary material to enhance accessibility for researchers and clinicians interested in employing this tool in their practice.
‘’ The adapted Richards-Campbell Sleep Questionnaire may also be utilized in various clinical settings beyond the intensive care unit, including general medical wards, palliative care, and outpatient consultations, thereby providing valuable assessments of patients' sleep quality in diverse situations.’’
|
Reviewer 3 Report
Comments and Suggestions for Authors
The title of the paper is "Cross-cultural adaptation of the Richards-Campbell Sleep questionnaire for intensive care unit patients in Morocco: Reliability and validity assessment". However, only one paragraph is devoted to cross-cultural adaptation, in the methods section. Besides, the methodology for this is outdated. It is true that other authors have followed this same methodology lately, but this does not add any merit to this approach. Translation is now regulated by ISO stadards, which must be used. It is not appropriate to carry out translation work if not performed by professional translators. This is equivalent to provide medical care by people who do not belong to the medical profession.
There is no mention to cross-cultural adaptation in the results section, but it appears in the discussion. If the study is about cross-cultural adaptation, this should be developed and be the main focus of the whole paper.
In my opinion, this article is not about cross-cultural adaptation. Instead, it is about the reliability and validity assessment of a cross-culturally adapted questionnaire, regardless of the methodology chosen for this cross-cultural adaptation, which I have already critized. The paper should be refocused so that it remains logical and consistent.
Author Response
|
We sincerely appreciate your thorough review and insightful comments on our manuscript titled "Cross-cultural adaptation of the Richards-Campbell Sleep Questionnaire for intensive care unit patients in Morocco: Reliability and validity assessment." Your feedback has highlighted several critical areas for improvement, and we are committed to addressing them comprehensively.
Methodology Update: We appreciate your observation regarding the methodology employed for translation and adaptation. We would like to clarify that the translators involved in this process are professionals who are proficient in both the source language and the target language, specifically Moroccan dialectal Arabic. We have followed the steps outlined in the "Guidelines for the Process of Cross-Cultural Adaptation" by Beaton (2000) rigorously, ensuring that the adaptation process meets the highest standards of quality and accuracy.
Refocusing the Manuscript: In light of your feedback, we will reframe our manuscript to emphasize that while the study includes a reliability and validity assessment, the primary focus is indeed on the cross-cultural adaptation of the questionnaire.n this sense we propose this title: Translation and Validation of the Richards-Campbell Sleep Questionnaire for Intensive Care Unit Patients in Morocco: Reliability and Validity Assessment |